# Discovery of senolytics using machine learning

Vanessa Smer-Barreto [1,6] ✉, Andrea Quintanilla[2,6], Richard J. R. Elliott[1], John C. Dawson [1], Jiugeng Sun[3], Víctor M. Campa[2], Álvaro Lorente-Macías [1], Asier Unciti-Broceta [1], Neil O. Carragher[1], Juan Carlos Acosta [1,2] ✉ & Diego A. Oyarzún [3,4,5] ✉

Cellular senescence is a stress response involved in ageing and diverse disease processes including cancer, type-2 diabetes, osteoarthritis and viral infection. Despite growing interest in targeted elimination of senescent cells, only few senolytics are known due to the lack of well-characterised molecular targets. Here, we report the discovery of three senolytics using cost-effective machine learning algorithms trained solely on published data. We computationally screened various chemical libraries and validated the senolytic action of ginkgetin, periplocin and oleandrin in human cell lines under various modalities of senescence. The compounds have potency comparable to known senolytics, and we show that oleandrin has improved potency over its target as compared to best-in-class alternatives. Our approach led to several hundred-fold reduction in drug screening costs and demonstrates that artificial intelligence can take maximum advantage of small and heterogeneous drug screening data, paving the way for new open science approaches to early-stage drug discovery.

Senescence is a cellular state characterised by permanent cell cycle arrest, macromolecular damage and metabolic alterations[1]. The senescent phenotype can be triggered by multiple cellular and environmental stressors, including replicative exhaustion, oncogenic activation, chemotherapy, and radiation[2], and is known to have beneficial and deleterious effects on tissue microenvironment[3]. For example, senescence aids mammalian embryonic development, promotes wound healing and stemness[4,5], and is a potent tumour suppression mechanism that restrains the growth of cells in danger of malignant alterations[6]. Conversely, senescent cells also promote tumorigenesis and various age-related malignancies due to the secretion of a complex set of proteins known as the senescence-associated secretory phenotype (SASP)[7,8]. Besides their role in cancer and ageing[9], the senescent programme has been linked to adverse effects in a broad range of

conditions, including osteoporosis, osteoarthritis, pulmonary fibrosis, SARS-CoV-2 infection, hepatic steatosis, and neurodegeneration[10]. As a result, there is a growing interest in the discovery of new senolytics, i.e. therapeutic agents that selectively target senescent cells for elimination[10].

Senolytics have shown substantial promise in ameliorating symptoms of many conditions in mice[11–20] and removal of senescent cells has also been linked to some adverse effects due to blockage of their beneficial roles in processes such as wound healing and liver function[21,22]. Despite encouraging results, to date there are few known compounds with proven senolytic action[11–19,23–28], and only two compounds have shown efficacy in clinical trials (dasatinib and quercetin in combination therapy[29]). Some of the most scrutinised senolytics were identified by targeting anti-apoptotic proteins upregulated in

[1]Cancer Research UK Edinburgh Centre, MRC Institute of Genetics and Cancer, University of Edinburgh, Crewe Road, Edinburgh EH4 2XR, UK. [2]Instituto de Biomedicina y Biotecnología de Cantabria (IBBTEC), CSIC-Universidad de Cantabria-SODERCAN. C/ Albert Einstein 22, Santander 39011, Spain. [3]School of Informatics, University of Edinburgh, 10 Crichton St, Edinburgh EH8 9AB, UK. [4]School of Biological Sciences, University of Edinburgh, Max Born Crescent, Edinburgh EH9 3BF, UK. [5]The Alan Turing Institute, 96 Euston Road, London NW1 2DB, UK. [6]These authors contributed equally: Vanessa Smer-Barreto, Andrea Quintanilla. ✉e-mail: vanessa.smerbarreto@ed.ac.uk; juan.acosta@unican.es; d.oyarzun@ed.ac.uk

senescence, such as the Bcl-2 family inhibitors navitoclax[28] and ABT-737[16]. Other senolytics were discovered through panel screens[17] and, more recently, screens have identified cardiac glycosides (ouabain[13], digoxin[15]) and BET inhibitors (ARV825[30], JQ1[31]) as potent senolytic agents. A key challenge for senolytic therapies to succeed is that many such compounds display cell-type specific action. In addition, certain senolytics that work well for one cell-type are highly toxic against other non-senescent cell-types[19,24]. In the case of cancer therapies, most known senolytics target pathways that are mutated in cancer, which limits their applicability as therapeutic agents[32] and highlights the need to discover new senolytics that could be employed in therapy.

In the past decade, computational screens based on Artificial Intelligence (AI) have been widely adopted by industrial and academic laboratories due to their ability to detect hidden patterns in large collections of chemical data[33,34]. These AI-powered screens can narrow down the chemical search space and have found applications in a range of tasks such as bioactivity prediction[35], target identification[36–38], virtual drug screening[39,40], and drug repurposing[41,42]. Most recently, generative models have been employed to generate novel chemical structures with prescribed properties[43,44]. Such approaches typically employ a combination of molecular dynamics simulations and sophisticated computational pipelines to navigate the space of drug candidates[45,46]. Recent years have witnessed the adoption of machine learning models trained on molecular fingerprints or learned representations of chemical structures[39,47–51], and several of these methods depart from traditional target-oriented approaches to drug discovery in favour of target-agnostic strategies[52] that employ phenotypic readouts for model training[39]. Such target-agnostic approach offers new avenues to expand the range of chemical starting points in early phases of the drug discovery pipeline[53], and is particularly well suited for senolytics discovery given the poor grasp of the molecular pathways that control the senescent phenotype.

In the context of cellular senescence, various works have employed machine learning for discovery of geroprotectors[54], ageing-related compounds[49–51,55] and anti-senescence compounds via convolutional neural networks trained on morphological features[48]. Bioinformatics approaches have also aided target identification of senescence-related compounds, senolytics and anti-senescent compounds[18,28,56].

Here, we report the development and validation of a machine learning pipeline for the discovery of senolytics. We assembled a dataset mined from multiple sources[11–19,23–28], including academic publications and a commercial patent, and employed it to train machine learning models predictive of senolytic action. We computationally screened a library of more than 4000 compounds and identified a reduced set of 21 candidate hits for experimental validation. Our experimental screen in two model cell lines of oncogene- and therapy-induced senescence revealed senolytic activity of three compounds: ginkgetin, oleandrin and periplocin, with potencies and dose-responses comparable to known senolytics. We further show that oleandrin has greater potency and activity over its target (Na$^+$/K$^+$ ATPase) and its senolytic effector NOXA, as compared to known cardiac glycosides with senolytic action. Our work demonstrates that machine learning can take maximum advantage of published screening data to find new active therapeutic compounds, laying the methodological groundwork for a new open science approach to drug discovery and repurposing.

## Results

### Data assembly and quality control

We first assembled a dataset of senolytics (positives) and non-senolytics (negatives) for model training (Fig. 1a). To this end, we mined a panel of 58 senolytics reported in the literature, including compounds from various chemical families such as flavonoids, cardiac glycosides, and antibiotics with senolytic action. These compounds were selected on the basis that they can maintain a minimum of 60% viability in normal cells and eliminate senescent cells for at least one cell line, one concentration, and one strategy for induction of senescence. While relaxing these constraints would have helped us increase the number of positives for model training, we prioritised consistency over the number of senolytic compounds. Wherever possible, we cross-validated different studies, e.g. for widely reported senolytics such as navitoclax, digoxin, ouabain, ABT-737 and fisetin. The selected panel of positives includes compounds that target the senescent phenotype in a variety of cell types (Fig. 1b). Some of these compounds, e.g. ouabain[13], have wide-spectrum senolytic action, whereas others such as BIX-01294, limit their effect to specific conditions[13]. We combined these positives with another panel of 19 senolytics reported in a commercial patent[14]. The full list of positives was then merged with a large background of compounds assumed to lack senolytic action. This assumption was needed due to the lack of data on negative screening results in the literature. Since machine learning models can bias their predictions towards the training data they have been exposed to, we chose the negatives from two diverse chemical libraries, LOPAC-1280 and Prestwick FDA-approved-1280, which contain a wide range of FDA-approved or clinical-stage compounds.

The full dataset for model training contains 2523 compounds, including 58 positives (2.3%). We deliberately chose to overrepresent the negatives in the training data so as to reflect the low likelihood of a chemical structure being a senolytic (Fig. 1a). To convert the chemical structures into a numerical format for model training, we binarised each compound in the training library as 0 (negative) or 1 (positive), and computed 200 physicochemical descriptors with the RDKit package[57]. These descriptors include basic molecular properties such as maximum partial charge, molecular weight, and number of valence electrons, as well as structural properties such as the molecular connectivity Chi indexes, E-State topological parameters and Kappa shape indexes.

The positive compounds were mined from highly heterogeneous sources that utilised different cell lines, screening assays, and methods for induction of the senescent phenotype (Fig. 1b). This bears the risk of introducing bias in our models and limit their predictive power if specific chemical families are overrepresented in the training data. This is further exacerbated by the heavy imbalance between the number of positives and negatives included in the training data. We thus sought to carefully quantify the diversity of the 58 positives using the RDKit descriptors as feature vectors associated with each compound. To assess diversity of the training data, we examined the cluster structure of the positive compounds using three different methodologies (Fig. 1c–e). We first clustered the positives with the k-means algorithm[58] and the cosine distance between feature vectors; this analysis revealed an almost linear decrease in the k-means score with respect to the number of clusters (Fig. 1c). The lack of a clear "elbow" in the k-means score suggests poor data clustering and hence provides a qualitative indication of diversity in the training data. To determine the quality and consistency of these clusters, we computed the silhouette coefficients for all compounds and number of clusters (k)[58]; we found consistently low values for the silhouette coefficient averaged across all compounds, which further suggests little similarity among the senolytic compounds chosen for training.

As a separate check for the diversity of the training data, we built the Tanimoto distance graph for all senolytics employed for training and labelled each compound according to the source from which they were obtained (Fig. 1d). Nodes in the Tanimoto distance graph represent compounds, and two compounds are connected by an edge if they are sufficiently close in the chemical descriptor space. The structure of the resulting distance graph corroborates the finding that most senolytics are far apart in the descriptor space (median Tanimoto distance = 0.77; Fig. 1d inset), and thus tend to be highly dissimilar to each other. As a final check, we clustered the Tanimoto distance graph

using community detection, a type of clustering technique from network science[59] that does not require a priori specification of the number of clusters. We employed the popular Louvain algorithm[60], because of its computational efficiency and the inclusion of a resolution parameter (γ) for tuning the granularity of the resulting clusters; larger values of γ lead to more clusters and hence a more granular partition of the graph. For a wide sweep of the resolution parameter, we found an almost linear increase in the number of clusters, and two plateaus at $k = 5$ and $k = 6$ clusters (Fig. 1e); such plateaus suggest that the data may naturally cluster into five or six groups. We further investigated these clusters and reasoned that compounds may aggregate according to the source where they were mined from. To test this hypothesis, we quantified the similarity between the Louvain clusters and literature labels (Fig. 1b) using the adjusted Rand index (ARI), a score for comparing different clusterings that corrects for random group assignments[61]. We found low ARI scores (Fig. 1e) across

all cluster resolutions described by the γ parameter; moreover, the ARI scores showed pronounced troughs at the plateaus detected with the Louvain method (mean ARI < 0.05 for 100 runs of the clustering method), which we regarded as sufficient evidence that compounds do not cluster according to the source from which they were obtained.

## Predicting senolytic compounds by computational screen with machine learning

We next sought to train machine learning models on the assembled dataset, with the aim of using them to computationally screen chemical libraries and identify hits for experimental validation (Fig. 2a). To this end, we first performed a feature selection process on the full dataset to reduce the number of features for training, before any cross-validation or train-test split. Using a random forest model and the average reduction of Gini index as a measure of impurity, we identified a reduced set of 165 normalised features (Supplementary Fig. 1a). This

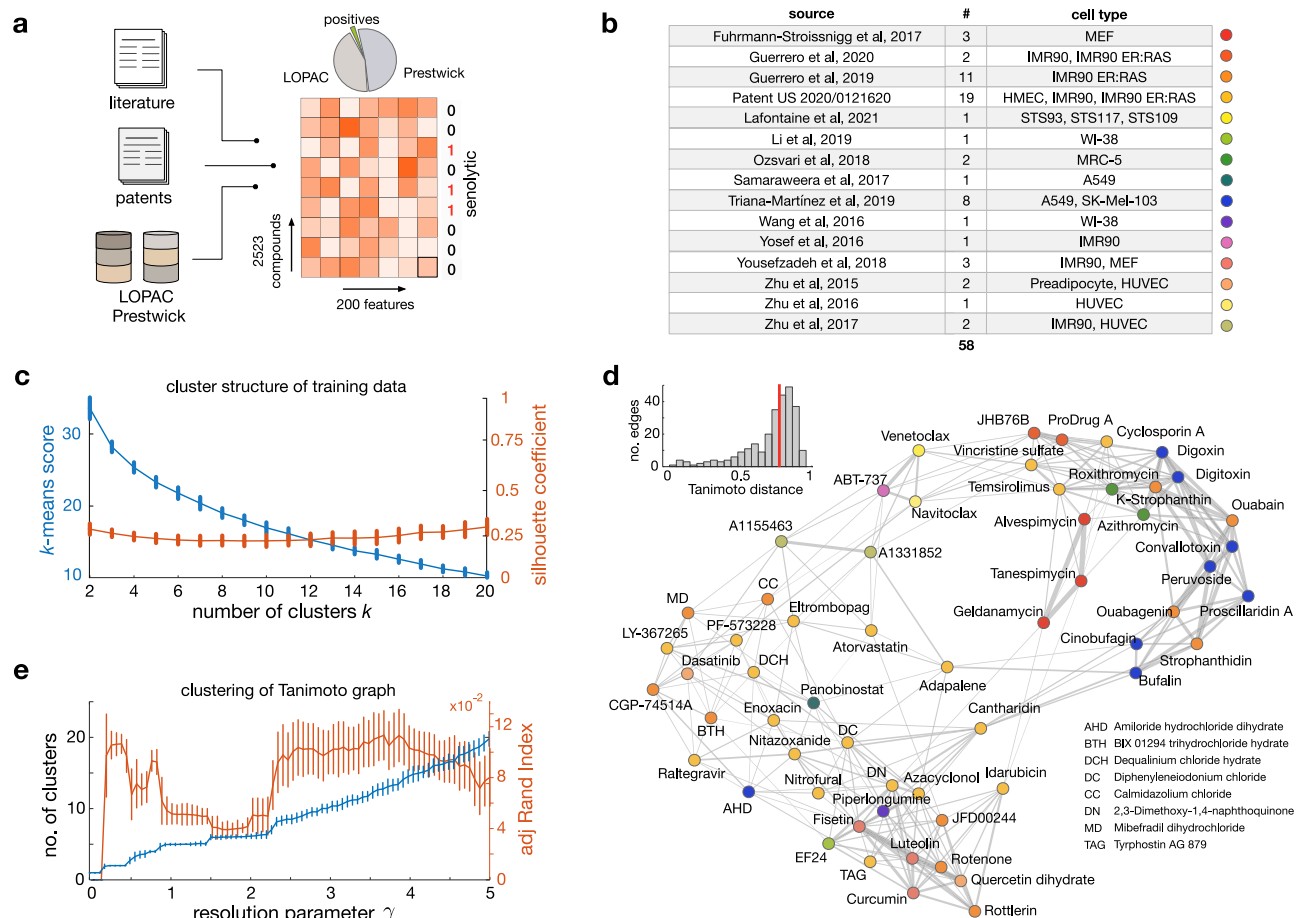

**Fig. 1 | Compounds employed to train machine learning models of senolytic action. a** We assembled training data from multiple sources. We mined 58 known senolytics (positives) from academic papers and a commercial patent, and integrated them with diverse compounds from the LOPAC-1280 and Prestwick FDA-approved-1280 chemical libraries (negatives). Chemical structures were featurised with 200 physicochemical descriptors computed with RDKit[57] and binary labelled according to their senolytic action. These labelled data were employed to train binary classifiers predictive of senolytic activity. **b** Sources of the 58 senolytics employed for training, including the number of compounds per source and the cell lines where senolysis was identified. **c** Cluster structure of the senolytics employed for training using the RDKit descriptors as features. Plot shows the k-means clustering score and silhouette coefficient[58] averaged across compounds for an increasing number of clusters (k). Error bars denote one standard deviation over 100 repeats with different initial seeds. The lack of a clear "elbow" in the k-means score and low silhouette coefficients suggest poor clustering among the senolytics

employed for training. **d** Tanimoto distance graph of senolytics employed for training; nodes are compounds and edges represent compounds that are sufficiently close in the physicochemical feature space. Node colour indicates the data source as in panel b. To emphasise the overall dissimilarity between compounds, we set the edge thickness as the Tanimoto similarity (1-distance). Inset shows the distribution of Tanimoto distances across the 269 graph edges (median distance of 0.77). **e** Clustering of the Tanimoto distance graph using the Louvain algorithm for community detection[60]. Plot shows the average number of clusters with respect to the resolution parameter (γ) across 100 runs (error bars denote one standard deviation); increasing values of γ produce a larger number of clusters. We observe pronounced plateaus at 5 and 6 clusters, suggesting some degree of clustering in the data. We computed the adjusted Rand index[61] (ARI) averaged across all compounds to quantify the similarity between cluster labels and compound source labels (15 labels; panel e). Low ARI values indicate that Louvain clusters are substantially different from the literature source labels.

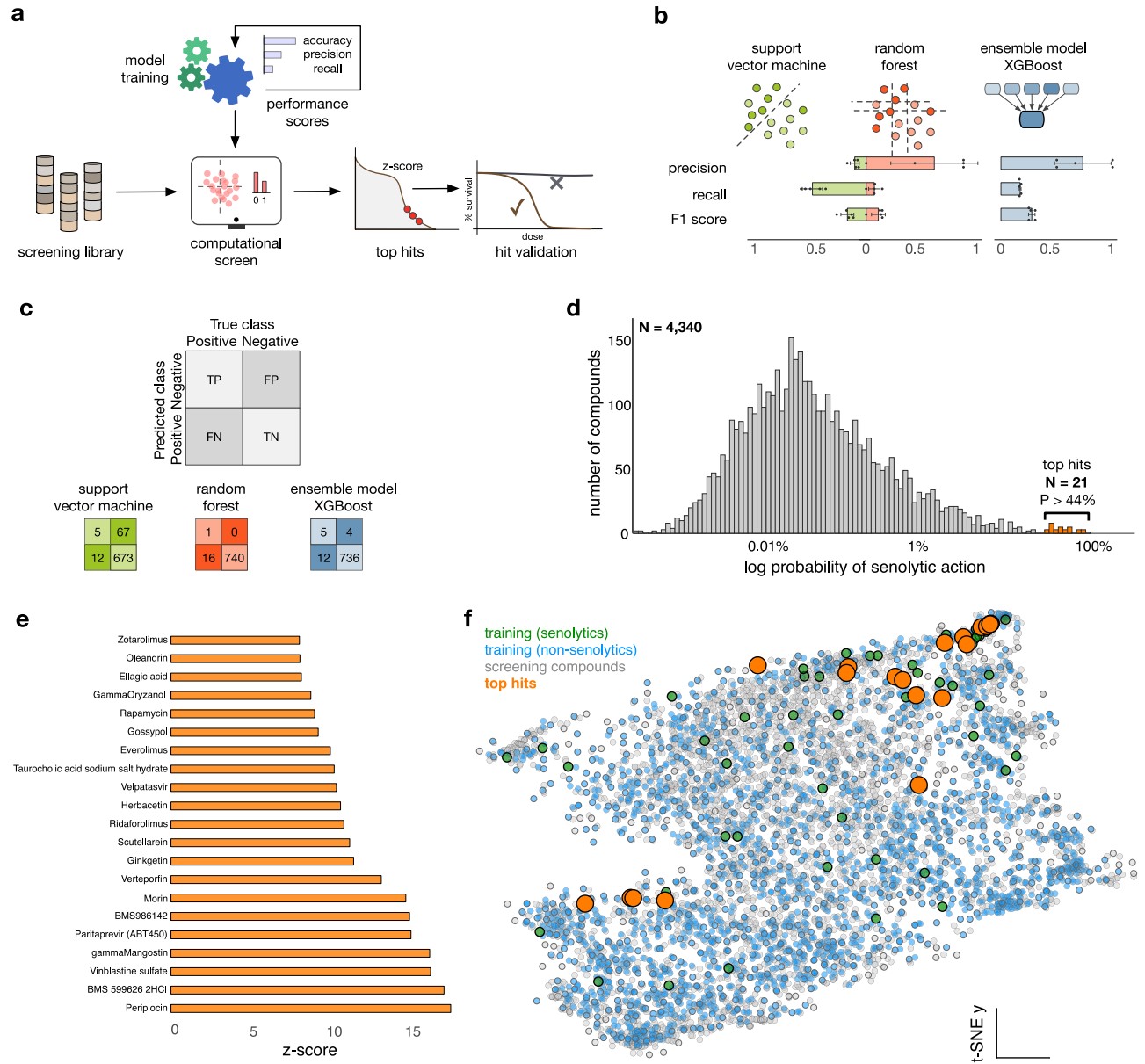

**Fig. 2 | Training of machine learning models and computational screening.**
**a** Pipeline for model training, compound screening, and hit validation. Several classification scores were used as performance metrics to determine the most suitable model for the computational screen. **b** Results from three machine learning models trained on 2523 compounds (Fig. 1a) and a reduced set of 165 features (Supplementary Fig. 1a); bar plots show average performance metrics computed in 5-fold cross-validation, with error bars denoting one standard deviation across folds. Mean ± s.d. are shown from $n = 5$ data folds. **c** The confusion matrices were computed from models trained on 70% of compounds, and tested on 17 positives and 740 negatives that were held-out from training. All models displayed poor performance metrics (Supplementary Table 1), and we chose the XGBoost algorithm for screening because of its lower number of false positives.

**d** Results from computational screen of the L2100 TargetMol Anticancer and L3800 Selleck FDA-approved & Passed Phase chemical libraries, totalling 4340 compounds. The XGBoost model is highly selective and scored most compounds with a low probability of having senolytic action; a small fraction of $N = 21$ compounds were scored with $P > 44\%$, which we selected for experimental validation. **e** Compounds selected for screening, ranked according to their $z$-score normalised prediction scores from the XGBoost model; the selected compounds are far outliers in the distribution of panel **c**. **f** Two-dimensional t-SNE visualisation of all compounds employed in this work; t-SNE plots were generated with perplexity 50, learning rate 200, and maximal number of iterations 1200[65]. Compounds with prediction scores above 44% from the XGBoost model are marked with orange circles.

relatively small reduction can be explained by the inherent high-dimensionality of the training data; a principal component analysis shows that more than 100 dimensions are needed to accurately explain the variability in the data (111 features for 99% of explained variance, Supplementary Fig. 1b). The selected 165 features were then utilised with the whole set of instances in the dataset (LOPAC, Prestwick, external sources) to train various models for binary classification of senolytics (Fig. 2b and Supplementary Table 2). For model selection we

performed 5-fold cross-validation on the whole dataset for fair comparison across models and to take full advantage of the limited number of positive samples. Due to the severe imbalance between the number of senolytic and non-senolytic compounds, we scored the models with three performance metrics: precision (fraction of true positive identifications out of all positive identifications), recall (fraction of correct identifications of true positives) and $F_1$ score (harmonic mean of the precision and recall). We note that model accuracy (fraction of overall

correct classifications), a common metric employed to score classification algorithms, is generally unsuited for imbalanced problems because it tends to produce overoptimistic results due to correct classification of the majority class, even when the minority class is poorly classified[62].

We focused primarily on two common models for binary classification: support vector machines (SVM) and random forests (RF). These models operate by partitioning the feature space so as to ensure that positive and negative samples are optimally assigned to a partition. In their basic form, SVM slices the feature space with a hyperplane, while RF are ensembles of decision trees that segment the feature space with orthogonal cuts across each feature[58]. We found that both SVM and RF models displayed poor performance (Fig. 2b) and showed marked differences in the type of misclassification errors they produce. The performance metrics (Fig. 2b) suggest that the RF model tends to return few false positives (high precision) and a high number of false negatives (low recall), whereas the SVM returns opposite results. We also evaluated a number of alternative models of varied complexity, including logistic regressors, a Naïve Bayes classifier, as well as data augmentation methods for imbalanced classification (SMOTE[63]); these additional models displayed worse performance than the SVM and RF models (Supplementary Table 2). The hyperparameters of the SVM and RF models were determined using 5-fold cross-validation (Supplementary Tables 3–4).

For the purposes of early-stage drug discovery, false positives are more deleterious than false negatives because they artificially inflate the number of predicted hits and thus increase the costs of downstream experimental validation. We thus took the performance of the RF model as a baseline, and aimed at improving its predictive power with an ensemble model (XGBoost) that is known to improve performance by aggregating predictions from a collection of decision trees[64]. The XGBoost model improved precision, recall, and $F_1$ scores, and overall returned the best performance among all considered models (Fig. 2b). We observed an average precision score of $0.7 \pm 0.16$ in 5-fold cross-validation on the whole dataset analysis of this model, which amounts to an average false discovery rate of 30% that we regarded as acceptable given the heterogeneity of the data employed for training. The hyperparameters of the XGBoost model were determined via 5-fold cross-validation (Supplementary Table 5). We also benchmarked the XGBoost model against a deep learning model based on message-passing neural networks that have shown excellent performance across a range of molecular property prediction tasks[39], but these were substantially outperformed by the XGBoost model (Supplementary Fig. 2). We re-trained the SVM, RF and XGBoost models on a stratified split (165 features, 70% for training, 30% for testing) to produce confusion matrices on the test set (Fig. 2c). These exemplify the trade-off of few false positives (high precision) and many false negatives (low recall) in the RF and XGBoost models, with opposite results for the SVM model.

We employed this final XGBoost model trained on 70% of data to screen a library of chemical structures designed on the basis of their diversity. We assembled compounds from the L2100 TargetMol Anticancer and L3800 Selleck FDA-approved & Passed Phase libraries into a single dataset with 4340 structures featurised with the physicochemical descriptors from RDKit; none of the compounds in the screening library were present in the training library. The XGBoost model proved to be exceptionally selective and produced a long-tailed distribution of prediction scores (Fig. 2d); most compounds were assigned extremely low prediction scores, and thus deemed to have a low probability of being senolytic. At the far end of the tail, the score distribution revealed a small group of 21 compounds (0.4% of the full library) with a comparatively higher probability of being senolytic ($P > 44\%$, Fig. 2d, orange), which we selected for further experimental validation. The selected compounds are extreme outliers, with prediction scores at least 8 standard deviations away from the bulk of

compounds screened (Fig. 2e). We employed dimensionality reduction to visualise the training and screening compounds in the RDKit feature space[65], which revealed a strong overlap between the two sets and thus a strong evidence that the computational screen was performed in the high-confidence domain of the machine learning model (Fig. 2f). The majority of the selected compounds are structurally diverse natural products, but with some common features including steroid saponins, flavone derivatives, and macrocycles.

We note that our feature selection process accessed the full dataset, and the best algorithm and hyperparameter settings were selected by applying a 5-fold cross-validation on the full dataset, too. Therefore, the performance metrics computed on a set of 30% held-out compounds are over-optimistic, since model selection benefitted from accessing those testing samples during feature selection and cross-validation on the full data. This could be avoided by applying feature selection and 5-fold cross-validation on the 70% training samples, but this would have reduced the training set to only 41 positive compounds, and thus make the selection of the best model less reliable. Since our aim was to employ the model to screen for new senolytics, we prioritised a reliable selection of the best model, at the risk of producing over-optimistic performance metrics.

## Identification of senolytics by experimental screening of top predicted compounds

We experimentally screened the top predicted molecules (Fig. 2e) for senolytic activity in two model cell lines for oncogene-induced and therapy-induced senescence. We first assessed oncogene-induced senescence (OIS) in human diploid fibroblasts IMR90 transduced with the fusion protein ER:RAS (IMR90 ER:RAS), which induces oncogenic Ras$^{G12V}$-mediated stress by addition of 4-hydroxytamoxifen (4-OHT) to the culture media[66]. Treatment of IMR90 ER:RAS cells with 4-OHT showed a decrease in proliferation, increased senescence-associated β-galactosidase activity, induction of cell cycle inhibitor expression, and activation of the SASP when compared with control and 4-OHT untreated cells, indicating that the cells underwent OIS (Supplementary Fig. 3).

To test for senolytic activity, we compared the effect of each compound on the total cell number (automated high content image-based analysis of total number of nuclei per well) in non-senescent and senescent IMR90 ER:RAS cultures treated with the top hits from the computational screen (Figs. 2e and 3a–d and Supplementary Fig. 4a–c). The drop in cell number compared to untreated controls is indicated by the nuclei count and reflects cell death[13,15]. As positive control, we employed ouabain, a cardiac glycoside with well characterised senolytic activity[13]. An optimal senolytic effect was reached with addition of 46.4 nM ouabain (IC50 control = 231 nM; IC50 senescence = 28 nM) to 4-OHT-induced IMR90 ER:RAS cells. This concentration killed most of the cultured senescent cells, but resulted in marginal reduction of the number of non-senescent control cells (Fig. 3b).

We then tested the 21 candidate compounds and found three with clear senolytic action: periplocin and oleandrin, two cardiac glycosides which have not been previously identified as senolytics, and ginkgetin, a natural non-toxic biflavone; this amounts to a hit confirmation rate of 14.28% (Fig. 3c, d and Supplementary Fig. 4c). Treatment of senescent IMR90 ER:RAS cells with the three compounds showed reduced nuclei counts when compared with proliferating, non-senescent IMR90 controls with an effect comparable to the positive control (Fig. 3c, d and Supplementary Fig. 5a). For confirmation, we performed cell staining with Hoechst to label the nuclei, showing the clearest effect with doses of 21.5 nM oleandrin (IC50 control = 85 nM; IC50 senescence = 14 nM), 46.4 nM periplocin (IC50 control = 300 nM; IC50 senescence = 24 nM) and 4.6 μM ginkgetin (IC50 control = 26 μM; IC50 senescence = 2.6 μM). These concentrations have a marginal effect on the nuclei counts in normal cells, but a marked decrease in nuclei counts in senescent cells (Fig. 3c, d and Supplementary Fig. 5a).

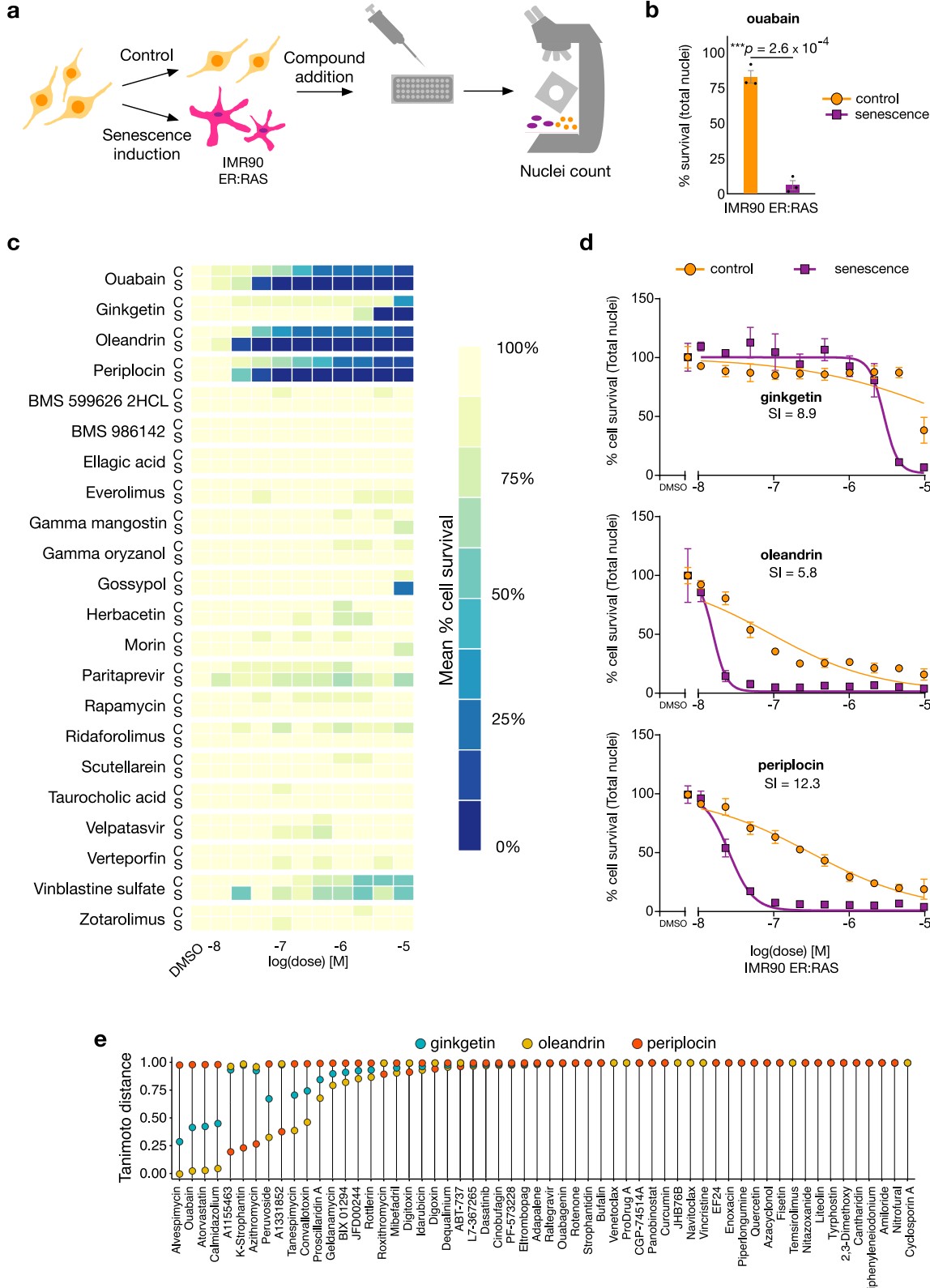

The dose-response curves showed a lower IC50 for the three compounds in nuclei counts from senescent cells as compared to nuclei from normal cells (Fig. 3d). In particular, periplocin (the top hit predicted by our model, Fig. 2d) showed a senolytic effect with a striking resemblance to the positive control ouabain (Fig. 3b, d).

We performed a second validation of the effectiveness of our machine learning models using a different stressor. We focused on therapy-induced senescence (TIS, Supplementary Fig. 6a), where human epithelial cancerous cells (A549) were induced to become senescent by addition of etoposide[66]. Cells were treated with 100 μM etoposide for 48 h, followed by another three days of exposure to media in standard conditions. As a positive control, we employed navitoclax (IC50 control = 10.2 μM; IC50 senescence = 440 nM), a Bcl-2 family inhibitor with well characterised senolytic activity[28]. Addition of

**Fig. 3 | Experimental characterisation of compounds selected for screening in oncogene-induced senescent (OIS) cells. a** Experimental setup of OIS model with IMR90 ER:RAS cells. Senescence was induced by addition of 4-OHT at 100 nM during the duration of the experiment (8 days). Control and senescent cells were plated in a 384-well plate on day five of 4-OHT induction. Top predicted compounds were added after multiwell seeding, and 72 h afterwards, the cells were fixed, and the nuclei stained and counted. **b** Bar plot of OIS positive experimental control, ouabain, at 46.4 nM. Data is normalised to DMSO. Data represented as individual points, and bars and error bars represent the mean ± SEM of three independent experiments. Statistical analysis was performed using a two-sided two-sample $t$-test for difference in mean value: ***$p < 0.001$; $p = 2.6 \times 10^{-4}$. **c** Results from experimental validation of controls and the top 21 compounds from Fig. 2d predicted to have senolytic action with $P > 44\%$. Three compounds out of the 21

displayed senolytic activity: ginkgetin, oleandrin and periplocin; heatmap shows mean across $n = 3$ replicates. This drug screen was done once with three experimental replicates. **d** Dose-response curves of the three newly found senolytic compounds. The senolytic index (SI) is defined as the ratio between the IC50 of control cells and the IC50 of senescent cells. Data is normalised to DMSO. Mean ± s.d. are shown from $n = 3$ experiments. Oleandrin and periplocin are related steroid saponins, similar to ouabain. Ginkgetin is a structurally distinct biflavone; the structures of the three compounds can be found in Supplementary Fig. 11. **e** Tanimoto distance between the three validated senolytics and those employed for model training; distances were calculated using the RDKit descriptors that were employed in the training of machine learning models in Fig. 2b and Supplementary Table 2. Source data are provided as a Source Data file.

1 µM navitoclax to A549 cells killed most of the cultured senescent cells, but resulted in no reduction in numbers of non-senescent control cells, confirming its optimal senolytic activity (Supplementary Fig. 6b).

Cells were then treated with the top 21 hits in Fig. 2d from our computational screen (Supplementary Figs. 6a–d and 7); we found that the same three compounds validated in oncogene-induced senescent cells (periplocin, oleandrin, ginkgetin) also displayed strong senolytic action in senescent A549 cells. The three compounds showed enhanced toxicity when compared with proliferating, non-senescent A549 controls with an effect comparable to the positive control. Hoechst labelling showed that a senolytic effect was reached with concentrations of 10 nM oleandrin (IC50 control = 19.5 nM; IC50 senescence = 5.4 nM), 46.4 nM periplocin (IC50 control = 267 nM; IC50 senescence = 72.2 nM) and 4.64 µM ginkgetin (IC50 control = 10.4 µM; IC50 senescence = 5.7 µM). These doses had a marginal effect on normal cells, but a decrease in survival rate in senescent cells (Supplementary Figs. 5b and 6c, d). The dose-response curves showed a lowered IC50 for the three compounds in senescent cells as compared to normal cells (Supplementary Fig. 6d).

To assess the chemical similarity between the three compounds correctly identified by the XGBoost model (Fig. 3d) and the senolytics employed for training (Fig. 2b), we computed the Tanimoto distance in the descriptor space between ginkgetin, oleandrin, and periplocin and each of the 58 senolytics in the training data (Fig. 3e). More than half of the training compounds were found to be maximally distant from our newly discovered senolytics, which provides some validation that our machine learning approach can effectively identify diverse compounds for specific biological effects such as senolysis. Both oleandrin and periplocin are steroid saponins, similar to ouabain, yet ginkgetin is a structurally distinct biflavone natural product. The steroid hormone core structure alone (Supplementary Fig. 8a, coloured red) present in periplocin, oleandrin and ouabain is insufficient for senolytic activity, when compared to inactive compounds taurocholic acid and gamma-oryzanol. Hence the glycoside linkages (Supplementary Fig. 8a, coloured blue) and furanone moiety (Supplementary Fig. 8a, purple) are likely contributing to target binding and potency, beyond physicochemical properties (e.g. solubility or cell permeability). Similarly, the basic flavone scaffold present in ginkgetin (Supplementary Fig. 8b, coloured blue) is also present in inactive compounds, herbacetin and morin. This may indicate a more complex target binding mode for the asymmetrical biflavone unit of ginkgetin and is worth further investigation. Several studies have shown that flavonoids are promising candidates in senescence-related research, hence the importance of finding new and more efficacious compounds of this kind[67–69].

### AI-identified compound oleandrin displays improved senolytic performance over benchmark senolytic cardiac glycosides

Our experimental screen suggested that oleandrin had enhanced senolytic activity as compared to the known cardiac glycoside ouabain, particularly in the low nanomolar range (Fig. 3c). We thus sought to compare the potency and mechanism of action of oleandrin to other

benchmark senolytics. We first compared the senolytic activity of the newly identified senolytics periplocin and oleandrin against ouabain at a low concentration of 10 nM. While ouabain and periplocin showed no cytotoxic activity in IMR90-ER:STOP proliferating control cells nor in IMR90-ER:RAS cells undergoing OIS, oleandrin showed a significant drop in cellular content in OIS cell cultures at 10 nM, which is indicative of an enhanced and highly specific senolytic activity at a lower drug concentration (Fig. 4a, b). We then tested whether oleandrin harbours similar potency in replicative senescence in IMR90 cells and A549 lung cancer epithelial cells undergoing TIS with etoposide treatment (Supplementary Fig. 9b, d). In both models, oleandrin was the only compound showing a significant reduction in cellular content in senescent cultures at a concentration of 10 nM, while keeping the cellular density unchanged in non-senescent control cell cultures. This indicates that oleandrin has a more substantial senolytic effect than ouabain in replicative senescence and therapy-induced senescence in epithelial cancer cells (Fig. 4c and Supplementary Fig. 9b, d, e). Importantly, treatment of oncogene-induced and replicative senescent cells with 10 nM oleandrin induced an evident increase in caspase-3/7 activity when compared to control cells, as well as cells treated with oleandrin and periplocin, confirming that oleandrin induces apoptosis in senescent cells at lower concentrations than ouabain (Fig. 4d and Supplementary Fig. 9f).

To further determine if oleandrin produces unwanted side effects in proliferation despite no cytotoxic effect in normal control cells, we subjected proliferating IMR90 cells to 10 nM ouabain, periplocin and oleandrin treatment, with etoposide as a control, and performed a proliferation assay by BrdU incorporation, observing that none of the cardiac glycosides induced unwanted changes in proliferation at 10 nM concentration (Supplementary Fig. 9g). The senolytic effect of cardiac glycosides has been previously linked to its canonical target, the Na$^+$/K$^+$ ATPase pump[13]. We confirmed that senescent cells displayed increased intracellular K$^+$. However, only oleandrin at 10 nM significantly reduced K$^+$ intracellular concentration during OIS and replicative senescence in IMR90 cells (Fig. 5a, b and Supplementary Fig. 9h, i), indicating that oleandrin inhibits its canonical target at lower concentrations than ouabain in senescent cells.

Senolytic cardiac glycosides activate a transcriptional programme resulting in the induction of several pro-apoptotic Bcl-2 family proteins[13]. Specifically, ouabain induces the expression of the BH3-only pro-apoptotic protein NOXA, which mediates its senolytic effect. We observed that only oleandrin induced a significant increase in NOXA mRNA expression at a low concentration of 10 nM (Fig. 5c, d), confirming the potent effect of oleandrin over benchmark cardiac glycosides. Moreover, after three days of treatment with 10 nM oleandrin, the mRNA expression of the senescence markers p16 and p21 was found to be reduced in the surviving cells in both OIS (Supplementary Fig. 9j) and replicative senescence (Supplementary Fig. 9k) cell cultures. Furthermore, treatment with oleandrin for three days resulted in reduced expression of proinflammatory cytokines IL1α, IL1β and IL8 mRNA in the surviving cells in OIS (Supplementary Fig. 9l) and

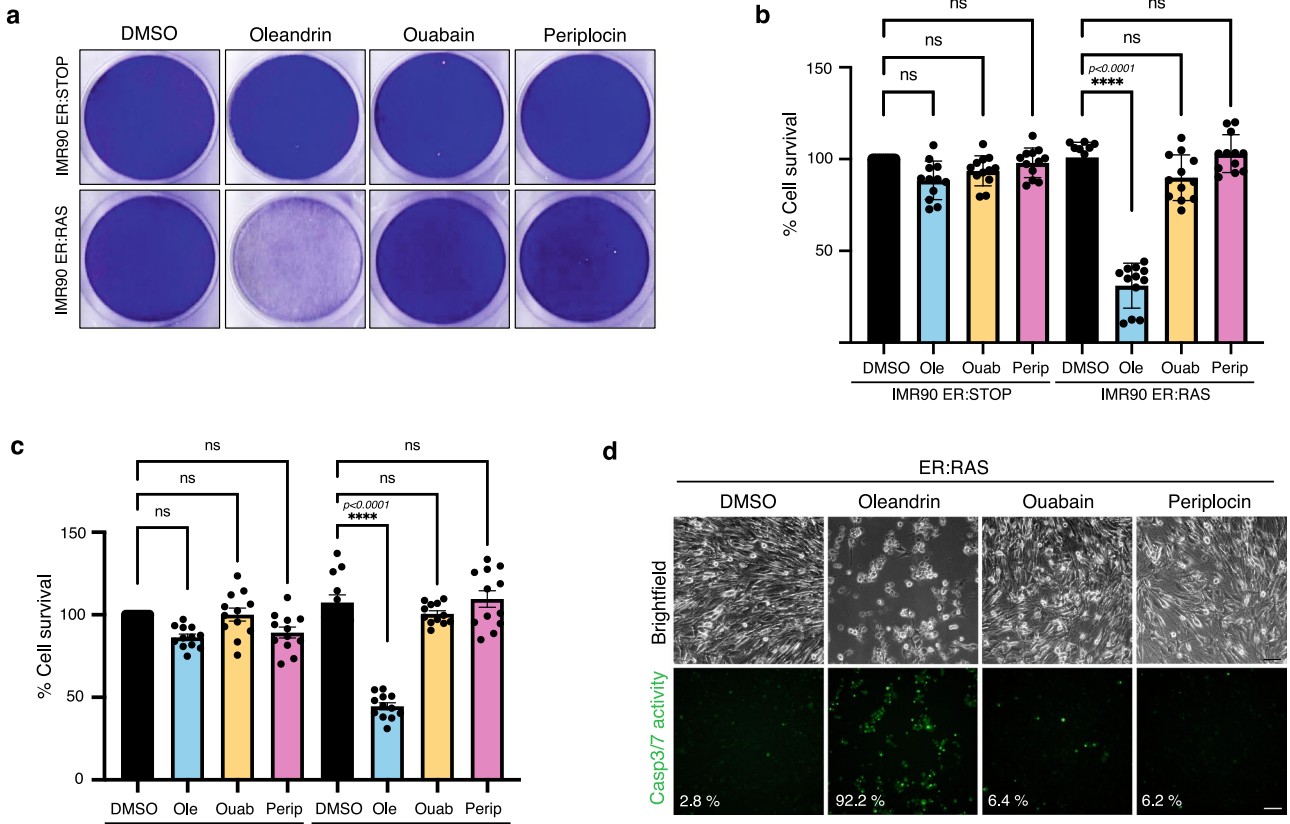

**Fig. 4 | Senolytic performance of oleandrin and periplocin. a** Cell survival assay measuring the senolytic effect in OIS. The panels show a representative crystal violet staining of tissue culture dishes of confluent senescent IMR90 ER:RAS and control IMR90 ER:STOP cells cultured with 100 nM 4OHT, and treated with 10 nM oleandrin, ouabain and periplocin, and DMSO as vehicle control for 72 h. **b** Cell survival by quantification of the crystal violet staining of the experiment shown in **a**, as described in "Methods" section. Data represented as individual data points, and bars and error bars representing the mean ± SEM of 12 independent experiments. Statistical analysis was performed using a one-way ANOVA (Tukey's test) for multiple comparisons. **c** Cell survival assay measuring the senolytic effect in replicative senescence. Graphs representing the cell survival by quantification of the crystal violet staining of confluent cultures of IMR90 cells at passage 27 (replicative senescence) and IMR90 cells at passage 13 (control) treated with 10 nM oleandrin, ouabain and periplocin, and DMSO as vehicle control for 72 h (related to Supplementary Fig. 9b). Data represented as individual data points, and bars and error bars representing the mean ± SEM of 12 independent experiments. Statistical analysis was performed using a one-way ANOVA (Tukey's test) for multiple comparisons. **d** Caspase 3/7 activity assay in control IMR90 ER:STOP and senescent IMR90 ER:RAS cells cultured in media containing 100 nM 4OHT, and treated during 35 h with 10 nM oleandrin, ouabain and periplocin, and DMSO as vehicle control. The panels show representative fluorescent images of caspase 3/7 positive cells (lower panels) and brightfield images (upper panels) of the same field for cell scoring. Percentage of green fluorescent cells per condition is indicated in the panel figures. Representative data of one of two independent experiments. Scale bars represent 100 μm. ns not significant, *$p < 0.05$, **$p < 0.01$, ***$p < 0.001$, ****$p < 0.0001$. Source data are provided as a Source Data file.

replicative senescence (Supplementary Fig. 9m). These findings provide evidence that oleandrin treatment reduced both senescence burden and proinflammatory SASP signalling. Altogether, these results demonstrate that our machine learning approach was able to discover an improved and more potent cardiac glycoside with senolytic action, facilitating the identification of chemical structure baits for further downstream chemical optimisation.

## Discussion

Current approaches to drug discovery suffer from notoriously high attrition rates in late-stage preclinical and clinical development. Due to their ability to parse and detect patterns in large volumes of data, AI has found applications across every stage of the drug discovery pipeline[70]. In this paper, we described a successful machine learning approach designed to identify novel drug candidates in early phases of the discovery process. We focused on targeted elimination of senescent cells, a phenotype that has attracted substantial interest for adjuvant cancer therapy[2], but for which few molecular targets have been identified. Our strategy revealed three compounds (ginkgetin, oleandrin and periplocin) that selectively eliminate cells displaying

oncogene- and therapy-induced senescence. We showed that these compounds have a potency comparable or higher to senolytics previously described in the literature and, crucially, our method led to large gains in efficiency by reducing the number of compounds for experimental screening by more than 200-fold.

Our approach offers several innovations that depart from current practice in AI for drug discovery. First, it relies solely on published data for model training, and thus avoids the extra costs for in-house experimental characterisation of training compounds. Second, our machine learning models were trained on just 58 chemical structures with proven senolytic action, which is much smaller data than typically considered in the field; the small number of senolytics in the training data is a consequence of senolysis being a rare molecular property and the limited number of senolytics reported in the literature so far. The success of our approach demonstrates that machine learning can take maximum advantage of literature data, even when such data is heterogeneous and of much smaller scale than typically expected[71]. Third, our models were trained in a target-agnostic manner using phenotypic signatures of drug action. Target specificity is of key importance for drug efficacy and safety in later stages of the discovery pipeline, but

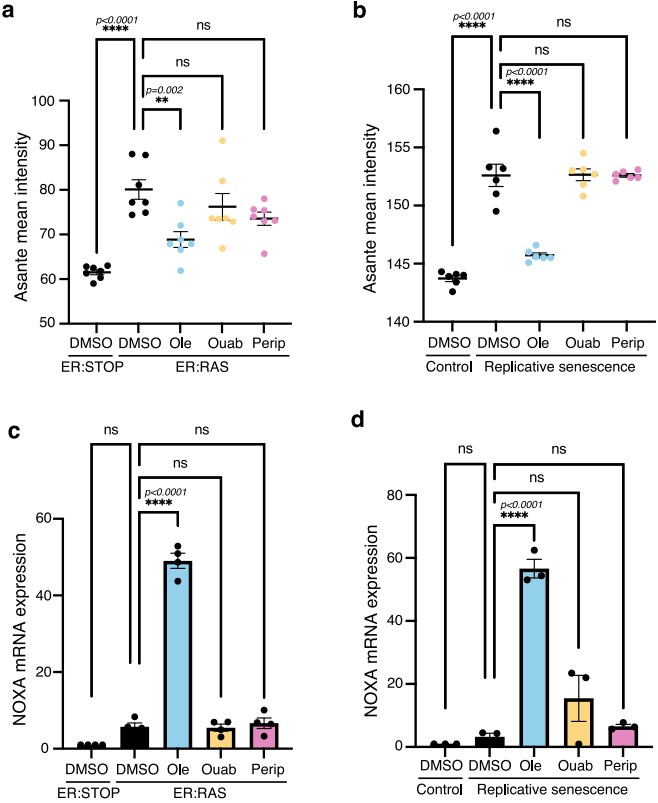

**Fig. 5 | Activity of oleandrin and periplocin on their senolytic targets.**
**a**, **b** Intracellular $K^+$ levels measured using Asante staining in **a** 100 nM 4OHT containing cultures of senescent IMR90 ER:RAS cells treated with 10 nM oleandrin, ouabain and periplocin, or DMSO as vehicle control, compared with IMR90 ER:STOP controls ($n = 7$), and **b** in IMR90 cells at passage 27 (replicative senescence) treated with 10 nM oleandrin, ouabain and periplocin, or DMSO as control vehicle compared to IMR90 cells at passage 13 (control) ($n = 6$). Data represented as individual data points and the mean ± SEM. Statistical analysis was performed using a one-way ANOVA (Tukey's test) for multiple comparisons. Representative images of Asante cell staining are shown in Supplementary Fig. 9h, i. **c**, **d** mRNA expression of NOXA determined by RT-qPCR in **c** 100 nM 4OHT containing cultures of senescent IMR90 ER:RAS cells treated with 10 nM oleandrin, ouabain and periplocin, and DMSO as control, compared to control IMR90 ER:STOP cells ($n = 4$), and **d** IMR90 cells at passage 27 (replicative senescence) treated with 10 nM oleandrin, ouabain and periplocin, and DMSO control compared to IMR90 proliferating cells at passage 13 (control) ($n = 3$). Data represented as individual data points and the mean ± SEM. Statistical analysis was performed using a one-way ANOVA (Dunnett's test) for multiple comparisons. ns not significant, *$p < 0.05$, **$p < 0.01$, ***$p < 0.001$, ****$p < 0.0001$. Source data are provided as a Source Data file.

there are numerous conditions of high economic and societal burden with few or no known targets[53]; for such conditions, there is an opportunity for phenotypic drug discovery to increase the number of chemical starting points that can be carried through the discovery pipeline[52].

A key challenge in computational drug screening is the construction of numerical representations of chemical structures that are predictive of drug efficacy[72]. With the advent of deep learning as the leading paradigm in the field[73], many recent works have developed such representations with e.g. transformer models for prediction of chemical reactions[74], graph neural networks to describe molecular structures[39,47], morphology-based convolutional neural networks for activity prediction[48] and generative models for de novo compound design[75]. In the ageing-related literature, previous studies have built pipelines to predict compounds that increase the life span of model organisms utilising chemical descriptors and gene ontology terms as features to train random forests[49], feature selection pre-processing to

train RF, SVM and neural networks[51], and molecular fingerprints to train RF models[50]. In our approach, we found that classic physico-chemical descriptors[57] calculated from SMILES strings were sufficient to train useful models. We observed limited benefits in the use of deep learning for compound featurisation, possibly because of the small size of our training data.

We found that careful data assembly, curation, and quality control were key for success. Since negative assays are rarely reported in the literature, we built the training data by pairing the known senolytics with a background of compounds assumed to lack senolytic action, but with an appropriate chemical diversity and a size deliberately chosen to reflect the paucity of senolytic compounds. These design choices produced a strong imbalance between the number of senolytic and non-senolytic compounds, which introduced additional challenges for model training. Several checks were needed to ensure that the training data was diverse enough and avoided bias toward specific chemical classes. Moreover, our models generally displayed poor performance as quantified by common classification metrics, producing large numbers of false positives and false negatives in cross-validation. We mitigated the impact of class imbalance by prioritising models with a lower number of false positives, and thus reduce the downstream costs for experimental validation. We carefully designed the screening library to balance similarity with the training data against exploration of novel chemical spaces (Fig. 2f). This led to an exceptionally selective distribution of prediction scores (Fig. 2d), which allowed us to select a cutoff for experimental validation with a reasonable number of hits and prediction scores far away from the bulk of the screening compounds. Although cutoff selection is highly problem-dependent, the robustness of results can be assessed with randomised repeats of model training and screening (Supplementary Fig. 10). Our results thus show that seemingly poor models can be employed effectively with adequate checks and balances on the structure of the data, plus a careful interpretation of misclassification errors.

Importantly, our approach identified oleandrin, a cardiac glycoside with stronger potency than the benchmark senolytic cardiac glycoside ouabain. Oleandrin has improved senolytic performance over ouabain, functioning at a low nanomolar range, inhibiting its canonical target and activating its senolytic pathway with higher efficacy. Moreover, we saw that oleandrin does not affect the proliferative capacity and viability of normal cells at that nanomolar concentration, indicating promising senolytic potential. Our work thus demonstrates that artificial intelligence and machine learning can help discover new and better-performing active compounds for a given pharmacological group. Further validation on animal models may strengthen the evidence for oleandrin as a promising new senolytic. A caveat, however, is that cardiac glycosides that have been employed in heart conditions have severe limitations due to toxicity[76], and our results suggest that oleandrin is not an exception because of its narrow therapeutic range and cardiotoxicity, and hence its use as systemic senolytic should be considered cautiously. The high potency of oleandrin could potentially benefit senolytic therapies administered locally on the site of damage; clinical trials are currently assessing such local administration of senolytics for osteoarthritis[10,77]. Moreover, in a separate work we have shown that ex-vivo senolytics perfusion of transplant discarded human livers preserves tissue architecture and its regenerative capacity during cold storage[78]. It is plausible that local oleandrin administration and perfusion in donor livers during the cold storage period before transplantation could overcome toxicity concerns from systemic administration and facilitate its use in the clinic.

From a translational point of view, we highlight that the three senolytics identified in this study are natural products found in traditional herbal medicines: *Ginkgo biloba* (ginkgetin)[79] *Nerium oleander* (oleandrin)[76] and *Periploca sepium* (periplocin)[80]. Oleandrin and

periplocin belong to the group of cardenolide glycosides, which are highly potent cardioactive agents, while ginkgetin is a biflavone with a broad pharmacological spectrum. Although they are not exempt of toxicological concerns, their already established ADME-Tox profiles in different models can help to reduce pharmacokinetics and tolerability issues during preclinical and clinical development. In principle, we do not rule out the senolytic potential of a low-toxicity compound like ginkgetin, but the exceptional activity and potency of oleandrin, together with its relatively low molecular weight (576.7 g/mol) and favourable cLogP (2.4), make it a promising lead candidate as compared to periplocin and ginkgetin (Supplementary Fig. 11). Oleandrin shares key structural features with other cardiac glycosides, including the presence of a sugar attached to the steroid core (at the C3β-OH group), a 2-furanone ring at C17β and an OH group at the C14β position of the steroid ring. Unlike most cardiac glycosides, oleandrin has an acetyloxy group attached at position C16β. In contrast to more structurally complex cardiac glycosides that display senolytic activity (e.g. ouabain, periplocin or lanatoside C), oleandrin features a monosaccharide and a simple central steroid system, which makes it closer to a potentially non-cardiotoxic pharmacophore and, consequently, an attractive starting point for future senolytic medicinal chemistry campaigns.

Our approach led to a significant reduction in experimental screening costs, largely because all models were trained solely on published data and, unlike other recent successes in the field[39], there was no need to screen compounds purposely for model training. The approach thus offers exciting prospects for new open science approaches to drug discovery. The COVID-19 pandemic spurred a multitude of such initiatives across the globe with the goal of finding new antivirals from the troves of published data[81]. Our work provides a concrete example of a simple yet effective machine learning pipeline that can be readily built from published screening data. We hope this approach will catalyse more open science approaches to discover treatments for conditions of unmet need, particularly those for which there is a limited grasp of the biological pathways involved in disease onset and progression.

## Methods

### A Data assembly, featurisation and quality control

**Training data.** We assembled a list of 58 previously identified senolytics mined from 15 sources[11–19,23–28] (Fig. 1b). The library of negative compounds contains 2465 compounds from the LOPAC-1280 (Library of Pharmacologically Active Compounds; Merck, Darmstadt, Germany) and Prestwick FDA-approved-1280 (Prestwick Chemical, Illkirch, France). We reasoned that these libraries are sufficiently diverse for training machine learning models. Although all compounds from these two library sources were assumed to be negative, it is plausible that some of these molecules were incorrectly labelled. This is because not all senolytics found have been expressly named in publications with screens (some sources only name a small set of their discoveries), or because these molecules have only been tested in several cell lines under one type of senescence induction.

**Featurisation.** The training dataset contains a string representation of the two-dimensional structures in the form of SMILES strings. The majority of SMILES were taken from the library of origin of every compound (LOPAC or Prestwick for training, Selleck or TargetMol for screening) with the exception of five positives (ProDrug A, JHB76B, CGP-74514A, A1331852, A1155463) whose SMILES were calculated using ChemDraw v18.1.0.535[12,13,19]. For chiral molecules, we favoured isomeric SMILES representations instead of the canonical case. We employed the RDKit package[57] to compute 200 physicochemical descriptions for each compound, which quantify different aspects of the molecular structure, its fragments, and global properties.

**Clustering analysis and Tanimoto graph.** To quantify the diversity of the senolytics employed for training (Fig. 1c–e), we performed $k$-means clustering of the 58 positives using the RDKit descriptors as features and the cosine distance function between $z$-score normalised feature vectors. The degree of clustering was quantified by the $k$-means score (Fig. 1c) defined as the within-cluster sums of point-to-centroid distances, summed across all clusters. The quality of the $k$-means clusters were determined with the silhouette coefficient $S$ averaged across the 58 senolytics. The silhouette coefficient varies between −1 and 1, with $S = 1$ indicating that compounds are in well separated clusters, $S = 0$ indicating overlapping clusters, and $S = −1$ indicating incorrect assignment of clusters. To build the Tanimoto distance graph (Fig. 1d) we first constructed a fully connected graph weighted by the pairwise Tanimoto distances feature vectors. This graph was then sparsified with the $k$-nearest neighbours graph ($k = 7$) intersected with the minimum spanning tree of the original graph. The edge widths were set as the Tanimoto similarity between compounds (1-distance). Clustering of the Tanimoto distance graph (Fig. 1e) was done with a Matlab R2022a implementation of the Louvain algorithm for community detection[60]. To compare the Louvain clusters with the compounds labelled according to their source (Fig. 1b), we employed the adjusted Rand Index which is a measure of the similarity between two clusterings, adjusted for the chance grouping of compounds[61]; low values of the ARI indicate little similarity between clusterings.

### B Model training and computational screen

All models were trained with the scikit-learn 0.24.1 library plus XGBoost 0.90[64] in Python. Models were trained on a reduced set of 165 $z$-score normalised features identified as relevant for classification using scikit-learn feature importance with a forest of trees function (Supplementary Fig. 1a) and the average reduction of Gini index as an impurity measure[82,83]. A PCA analysis was performed using the function prcomp from R's stats package 4.0.2 (Supplementary Fig. 1b). Dimensionality reduction of this data proved to be challenging, as more than 50% of the 200 RDKit features were needed to explain 99% of the variance.

For model selection, we trained on the whole set of data instances after feature reduction (165 columns, 2,523 rows) to perform fair comparisons across models and take full advantage of the limited number of positive samples. This was done with 5-fold cross-validation to check for overfitting. The metrics of this analysis are displayed in the bar charts of Fig. 2b and in Supplementary Table 2. All models were scored with three metrics of classification performance:

$$precision = (TP/(TP + FP)), \qquad (1)$$

$$recall = TP/(TP + FN), \qquad (2)$$

$$F_1 = TP/(TP + 1/2(FP + FN)), \qquad (3)$$

where TP, TN, FP and FN are the number of true positives, true negatives, false positives and false negatives, respectively. We attempted to resolve the severe class imbalance problem inherent to our data by utilising the pre-processing technique SMOTE on several classification algorithms, without significant improvement; SMOTE was applied to the training set only. For the SVM and RF, we tuned the 'class weight' parameter, which when set to 'balanced' rather than the default 'None', adjusts weights in a manner inversely proportional to class frequencies, therefore imposing heavier penalties in the misclassification of the less represented class. In the case of SVM, the setting of 'class weight' to 'balanced' was the best option, whereas for the RF the best setting was the default one. For the XGBoost model, we utilised this same data (165 columns, 2,523 rows) to optimise the hyperparameter 'max depth' using a grid search across all integer values in the interval

max_depth = [1,10] to maximise precision over 5-fold cross-validation runs; hyperparameter optimisation settings were applied only to the training set in each of the five folds. The hyperparameters of the SVM, RF and XGBoost models can be found in Supplementary Tables 3–5.

We subsequently re-trained the XGBoost, RF, and SVM models with a stratified random split of the data (70% for training, and 30% for testing) to produce the confusion matrices in Fig. 2c. For the computational screen, we employed the XGBoost model ran on the L2100 TargetMol Anticancer (TargetMol Chemicals, Wellesley Hills, MA) and L3800 Selleck FDA-approved & Passed Phase (Selleck Chemicals, Houston, TX) libraries. We used the XGBoost model trained on the 70:30 split to compute prediction scores on the screening library, i.e. the probability of a compound being classified as senolytic (Fig. 2d).

**C Validation assays**

We performed experimental validation of the compounds with z-score > 8 (Fig. 2e). These correspond to 21 compounds out of a total of 4340 compounds in our screening libraries. We employed two cellular models of senescence: one of OIS and a second one of TIS. For the OIS case, we utilised IMR90 ER:RAS cells with 4-OHT at 100 nM. The 4-OHT treatment had a duration of six days. For the TIS model, we used A549 cells with etoposide at 100 μM. The exposure of the cells to etoposide lasted 48 h, after which period the cells were cultivated for a further three days with normal media.

**Cell culture.** IMR90 (CCL-186) female human foetal lung fibroblasts and A549 (CCL-185) human lung adenocarcinoma cells were obtained from the American Type Culture Collection (ATCC, Manassas, VA). The cell lines were confirmed to be mycoplasma negative (Lonza MycoAlert, cat #LT07-118). IMR90 ER:RAS is a derivative of IMR90 cells expressing a switchable version of oncogenic H-Ras[84]. IMR90 ER:RAS and A549 cells were maintained in Dulbecco's Modified Eagle Medium (DMEM, ThermoFisher) supplemented with foetal bovine serum (10%), L-glutamine (2 mM, ThermoFisher), and antibiotic-antimycotic solution (1%, ThermoFisher) and incubated under standard tissue culture conditions (37 °C and 5% $CO_2$). For induction of the senescent phenotype, IMR90-ER:RAS cells were cultured in hydroxytamoxifen (4-OHT) (Sigma) added media at 100 nM final concentration. IMR90 were kept in culture for over 27 passages for replicative senescence. Cells were tested for mycoplasma on a regular basis.

**Quantification of senolytic action.** Cells were seeded (50 μL per well) into 384-well plates (IMR90 ER:RAS cells in Nunc Optical Bottom Polybase Microplates [#142761, Thermo Scientific, Rochester, NY] and A549 cells in CELLSTAR Cell Culture Microplates [#781091, Greiner Bio-One, Kremsmünster, Austria]). Cells were incubated under standard tissue culture conditions for 24 h before the addition of compounds. Passage 14 IMR90 ER:RAS cells were seeded at 1300 cells per well in the control condition, and at 1600 cells per well in the senescent case. Passage 34 A549 cells were seeded at 7000 cells per well in the control condition, and at 10,000 cells per well in the senescent case.

Dose response plates were prepared with a DMSO control and a 10-point half-log concentration range, and added to the compounds using a D300e digital dispenser (Tecan Trading AG, Switzerland) at a final concentration of between 10 μM and 10 nM. Every screened condition was carried out in triplicate. After 72 h of incubation with exposure to the compounds, cells were fixed by the addition of an equal volume of formaldehyde (8%, 50 μL; #BP531-500, Fisher Bioreagents, Fisher Scientific, Loughborough, Leicestershire) to the existing media, incubated at room temperature (30 min), and washed three times in phosphate-buffered saline (PBS). Cells were then permeabilised in Triton-X100 (0.1%, 50 μL) and incubated at room temperature (30 min) followed by three more washes with PBS.

Cells were stained with Hoechst 33342 for nuclei count (excitation/emission wavelength at 387/447 nM, DAPI channel, original concentration at 10 mg/ml, final concentration at 2 μg/ml; H1399, Molecular Probes, Eugene, OR). The staining solution was prepared in bovine serum albumin solution (10%). The staining solution was added to each well (30 μL) and incubated in the dark at room temperature (30 min), followed by three washes with PBS and no final aspiration. Plates were foil sealed.

**Image acquisition.** Plates were imaged on an ImageXpress micro XLS (Molecular Devices, Eugene, OR) equipped with a robotic plate loader (Scara4, PAA, UK). Four fields of view were captured per well (20x objective for A549 cells, 10x objective for IMR90 ER:RAS cells) and one filter was used (DAPI). A typical wild-type field of view contained 1000 cells in the IMR90 ER:RAS case, and 1400 in the A549 case.

**Image and data analysis.** The stained cell nuclei were counted on MetaXpress v6.6.2.46 software. The results per compound, phenotype condition, and dose were added and the results morphed into data frame format with functions from R's dplyr, tidyr, and reshape2 libraries.

The dose-response data (control plus 10 half-log range points) was fitted (ordinary least squares) to a log (inhibitor) vs normalised response (control value per condition [senescent, non-senescent] was constrained at 100%) with variable slope equation using Prism 6 software (GraphPad, San Diego, CA). With this fit, IC50 values were calculated for senolytic compounds.

**Compounds.** The following compounds were used in the present study: etoposide (Sigma-Aldrich, E1383), 4-hydroxytamoxifen (Sigma-Aldrich, H7904), ouabain (Apexbio, B2270), navitoclax (Apexbio, A3007), ginkgetin (Cayman Chemical, 25103-1 mg), oleandrin (Cayman Chemical, 29871-1 mg), periplocin (Cayman Chemical, 25216-1 mg), BMS 599626 dihydrochloride (Apexbio B5792), BMS 986142 (BioVision, B2420-1), ellagic acid (Apexbio, A2306), everolimus (Cayman Chemical, 22559-1 mg), herbacetin (Cayman Chemical 19285-1 mg), morin (MedChemExpress LLC, HY-N0621-10mg), paritaprevir (MedChemExpress HY-12594), rapamycin (Cayman Chemical, 13346-5 mg), taurocholic acid sodium salt hydrate (Selleck Chemicals, S5130), velpatasvir (BioVision, B1194-5), verteporfin (Apexbio, A8327), zotarolimus (Cayman Chemical, 29246-5 mg), gamma mangostin (MedChemExpress LLC, HY-N1957-5mg), gamma oryzanol (MedChemExpress LLC, HY-B2194), gossypol (MedChemExpress LLC, HY-15464A-10mg), ridaforolimus (Apexbio, B1639), scutellarein (MedChemExpress LLC, HY-N4314-1mg), vinblastine sulfate (MP Biomedicals LLC, 0219028725).

**Cell survival assay using crystal violet staining.** Equal numbers of control and senescent cells were plated at high density on multiwell plates just before senolytic treatment addition. After treatment, cells were fixed on 0.5% glutaraldehyde (Sigma) and then dried at RT overnight. Then, wells were stained with crystal violet solution for 3 h, washed and dried. Once they were completely dry, the plates were scanned for records. Quantification was performed extracting the staining with 1% acetic acid and measuring the final solution by absorbance at 595 nm.

**Cell proliferation assay using crystal violet staining.** 50,000 cells per 10 cm-plate were seeded and kept in culture. After 12 days, cells were fixed with 1% glutaraldehyde (Sigma) for 1 h. After several washes with water, dried plates were stained with 0.15% crystal violet solution for 1 h and then washed again. Once they were completely dried, plates were scanned for analysis.

**Caspase 3/7 activity assay.** IMR90 ER:STOP and IMR90 ER:RAS cells were cultured with 100 nM 4-OHT for 5 days. Cell were then plated in a multiwell plate at high density. Once cells were attached, Caspase 3/7 probe (C10423 Invitrogen) and senolytic drugs, oleandrin, ouabain and periplocin at 10 nM, were added to the media and cultured for 36 h. Time lapse imaging was recorded every hour on a Leica AF6500 microscope (10x). ImageJ 1.53k was used for positive cell counting.

**Intracellular K⁺ determination with Asante staining.** Cells were cultured for control and senescent phenotype and then treated with the senolytics oleandrin, ouabain and periplocin at 10 nM for 72 h. Asante (Asante Potassium Green-2 AM, abcam ab142806) was added to the culture 30 min before fixation, following manufacturer instructions. DAPI staining was done before image acquisition on Nikon TI2 microscope (20x). Image analysis was performed using ImageJ 1.53k for setting intensity threshold and measuring mean intensity.

**mRNA gene expression analysis.** Total RNA was isolated using RNAeasy kit (Quiagen). cDNA was generated with reverse transcriptase iScript (Bio-Rad). RT-qPCR was performed using SYBR Select Master Mix (Applied Biosystems) in OneStepPlus detection system (Applied Biosystems). Oligos for amplification were: NOXA Fw-CATGAGG GGACTCCTTCAAA and Rv-TTCCATCTTCCGTTTCCAAG; b-ACTIN Fw-CATGTACGTTGCTATCCAGGC and Rv-CTCCTTAATGTCACGCACGAT; IL1a Fw-AGTGCTGCTGAAGGAGATGCCTGA and Rv- CCCCTGCCAAGC ACACCCAGTA; IL1b Fw-TGCACGCTCCGGGACTCACA and Rv- CATGG AGAACACCACTTGTTGCTCC; IL8  Fw-GAGTGGACCACACTGCGCCA and Rv-TCCACAACCCTCTGCACCCAGT; p16 Fw-CGGTCGGAGGCCGA TCCAG and Rv- GCGCCGTGGAGCAGCAGCAGCT; p21Fw-CCTGTCAC TGTCTTGTACCCT and Rv- GCGTTTGGAGTGGTAGAAATCT.

**Western blot analysis.** Cell lysates were prepared using Cell Lysis Buffer (Cell Signalling 9803S). Clear lysates were quantified by Bradford colorimetric assay. Samples were resolved by polyacrylamide gel electrophoresis and transferred on nitrocellulose membrane, which was blocked by a 5% milk-TBS-Tween buffer for 1 h at RT. Primary antibodies (anti-p21 Sigma P1484, anti-p16 JC8, anti-IL1B MAB201 R&D) were incubated o/n at 4 °C. After 2 washes with TBS-Tween buffer, secondary antibodies were added for 1 h at RT and then 2 washes were done before developing by using enhanced chemical luminescence (Amersham) detection reagent. HRP-B-actin was incubated for normalisation.

**SA-β-galactosidase assay.** After 10 days in culture, cells were fixed in 0.5% glutaraldehyde (Sigma) for 10 min at RT. Then the cells were washed and stained with SA-β-Gal staining solution (20× KC [100 mM K3FE (CN)6 and 100 mM K4Fe(CN)6*3H2O in PBS], 20× X-Gal solution (Thermo Fisher Scientific) diluted to 1× in PBS/1 mM MgCl₂ pH 6). Staining was performed overnight at 37 °C in the dark. Once cells were washed, images were taken using an inverted tissue culture widefield microscope (Nikon) for documentation and quantification.

**Immunofluorescence and imaging.** IMR90 ER:STOP and IMR90 ER:RAS cells treated with 4-OHT during 8 days were fixed with 4% paraformaldehyde for 30 min. After several washes, cells were permeabilised with 0.2% Triton-100 for 10 min and then blocked for 30 min with PBS-BSA-Gelatin fish (Sigma). Primary antibodies (IL1A AF-200-NA R&D Systems; IL1B MAB201 R&D Systems and IL8 MAB208 R&D Systems) were prepared in a blocking buffer and incubated for 1 h at RT. Alexa Fluor secondary antibodies were used for signal detection and DAPI solution was added for 10 min. Finally, samples were washed before imaging. Confocal images (512 × 512 pixels; 0.76μm/pixel) were acquired sequentially on a SP5 laser-scan microscope (Leica) with a ×20 NA objective and 2× electronic zoom using LAS AF acquisition

software. Cells were excited sequentially with 405 nm and 594 nm laser lines and emission was captured between 430-480 nm (DAPI) and 605-655 nm (Alexa594) respectively. Images are presented after digital adjustment of curve levels to maximise signal with ImageJ. In all cases, exposure time, sensor gain, and digital manipulation were the same for control and experimental samples. Fluorochromes and colours are as indicated in the figure legends.

**BrdU incorporation assay.** 5-bromo-2′-deoxyuridine (BrdU) incorporation was used to measure the number of cells actively replicating DNA. Cells were incubated with 10 µM BrdU (85811, Sigma) for 18 h. After that, cells were fixed with 4% Paraformaldehyde, permeabilised, blocked and then stained for immunofluorescence using BrdU primary antibody (555627 BD Pharmigen), Alexa fluor secondary for detection, and DAPI staining for cell counting. Acquisition was done with a Nikon TI2 microscope and analysis was performed with ImageJ for cell counting.

**Data acquisition and statistical analyses.** For in vitro biological experiments, cell culture plates were randomly assigned to treatments in each experiment. Most imaging data were acquired and analysed automatically by a high content microscopy platform and the imaging analysis software MetaXpress, and thus, data collection and analysis were blinded. For all other experiments, data collection and analysis were blinded to the person collecting and analysing the data, and the samples were identified only at the end of each experimental analysis. All measurements were taken from distinct samples, as noted in figure legends, and no data were excluded. Sample sizes were based in standard protocols in the field. Unless otherwise stated, at least three biological independent replicates were performed for each experiment. Statistical analyses were performed using GraphPad Prism 9. All experimental replicates are plotted in the graphs as individual data points. All experimental data is available in the data source file. Statistical significance for each experiment was established by one-way ANOVA using the built-in tools of Prism 9. Statistical tests are indicated in the figure legends and were chosen based on the nature of the experiment and the standard tests employed in the field. Underlying assumptions for these tests, including sample independence, variance equality, and normality were assumed to be met although not explicitly examined. One-way ANOVA was followed by Tukey's or Dunnett's multiple comparison test when appropriate, as indicated in the figure captions. Two-sample statistical tests (Fig. 3b) were performed with a two-sided t-test with the R package rstatix 0.7.0. Asterisks denote $p$ value as follows: ns = not significant, $*p < 0.05$, $**p < 0.01$, $***p < 0.001$, $****p < 0.0001$.

### Reporting summary
Further information on research design is available in the Nature Portfolio Reporting Summary linked to this article.

### Data availability
Data for model training and computational screen is available at https://doi.org/10.5281/zenodo.7870357. Source data are provided with this paper.

### Code availability
Python code for model training and computational screening are available in Zenodo at https://doi.org/10.5281/zenodo.7870357. We employed Python v3.8.3 and the following packages: seaborn (0.10.0), numpy (1.18.1), pandas (1.0.1), matplotlib (3.1.3), sklearn (0.24.1), pickle (4.0), and xgboost (0.90).

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

## Acknowledgements

The authors disclosed receipt of the following financial support for the research, authorship, and/or publication of this article: V.S.B. is a cross-disciplinary post-doctoral fellow supported by the University of Edinburgh and the Medical Research Council (MC_UU_00009/2). J.C.A. acknowledges funding by Cancer Research UK (CRUK) (C47559/A16243 Training & Career Development Board - Career Development Fellowship), the University of Edinburgh Chancellor's Fellowship R42576 MRC, the Ministry of Science and Innovation of the Government of Spain (Proyecto PID2020-117860GB-I00 financed by MCIN/ AEI /10.13039/501100011033) and the Spanish National Research Council (CSIC). D.A.O. was supported by the United Kingdom Research and Innovation (grant EP/S02431X/1).

## Author contributions

V.S.B. designed research, performed data assembly, curation, computational analyses and experimental validation. A.Q. designed and performed in vitro research and experiments for senolytic potency and mechanism, performing characterisation of cellular senescence. A.Q. also provided counsel and training on experimental protocols. RJRE provided input on chemical analyses; R.J.R.E. and J.C.D. assisted with experimental screening, image analysis and data interpretation. J.S. assisted with model training. V.C. assisted with data acquisition and analysis. A.L.M. and A.U.-B. provided expertise in medicinal chemistry. N.O.C. provided expertise in drug discovery and high throughput screening. J.C.A. and A.Q. designed experimental work and provided expertise on cellular senescence. D.A.O. analysed data diversity and

provided expertise in machine learning. J.C.A. and D.A.O. provided overall supervision and direction of the work.

## Competing interests

The authors declare no competing interests.
