## [Peer Review File · Nature Communications]

Discovery of senolytics using machine learningEditorial Note: This manuscript has been previously reviewed at another journal that is not operating a transparent peer review scheme. This document only contains reviewer comments and rebuttal letters for versions considered at *Nature Communications*.

REVIEWER COMMENTS

Reviewer #1 (Remarks to the Author):

This revised version of the paper has a clearer and more detailed description of the machine learning methodology, by comparison with the original submission; and the authors have provided detailed responses to my previous comments. The main remaining issues are as follows.

Lines 192-195:

“For model selection we performed 5-fold cross-validation on the dataset (2,523 rows, 165 columns) to perform fair comparison across models and take full advantage of the limited number of positive samples.”

And from the authors’ responses to the reviewers’ comments:

“...the feature importance calculation was performed with the whole dataset, as a preliminary step before any train-test split.”

“In the revised Results section (Lines 182-253) we now explicitly detail the steps for model selection and model testing:

- The results in Fig. 2b and Supplementary Table 2 were computed using 5-fold cross-validation on 100% of the dataset and the reduced set of 165 features, to perform fair comparisons between all models and take full advantage of the limited number of positive samples.
- After we identified XGBoost as the best model on the five cross-validation runs, we retrained the three main models (XGBoost, SVM, RF) on a 70% split and tested them on a 30% held out set. These test results are reported in the confusion matrices of Fig. 2c. The XGBoost model retrained on this 70% split was the one employed for the computational screen.”

It is important to note that the above pieces of text mean that the predictive accuracy results reported for final models, trained on 70% of the data and evaluated on the 30% held out test set, are over-optimistic predictive accuracies. This is because the dataset which was split into 70% for training and 30% for testing is the same dataset which was used to perform feature selection and to select the best algorithms and best their hyperparameter settings using 5-fold cross-validation.

Note that, when performing 5-fold cross-validation on the full dataset, this means every sample (compound) is used exactly once (in one of the 5 iterations) as a test sample. In this case 5-fold cross-validation was used to evaluate several algorithms, each with different candidate hyperparameter settings, and each cross-validation run of a classification algorithm with a given candidate setting is effectively using every sample (compound) as a test sample once. In addition, feature selection was also performed using the full dataset.

Hence, when the same dataset is split into 70% for training and 30% for the final testing of the best models (with their best hyperparameter settings), the choice of those best models and their best hyperparameter settings already benefitted from previously accessing the full data during previous cross-validation runs, and also benefitted from feature selection using the full data. I.e., this final set of 30% of testing samples is not really a “proper test set” in a strict machine learning sense (it does not consist of previously unseen examples) because each of those samples already had its class predicted multiple times in multiple previous runs of the cross-validation procedure, and the results of those multiple predictions were used to select the best algorithms and their best hyperparameter settings.

The authors provided a justification for this experimental methodology, which was, quoting from lines

194-195:

“to perform fair comparison across models and take full advantage of the limited number of positive samples.”

This procedure indeed performs a fair comparison between the models, although this fair comparison could also be achieved by performing the feature selection process and the 5-fold cross-validation on the 70% of training samples rather than the full dataset, which would avoid the over-optimistic results and would be better from a strict machine learning perspective (but see below the discussion putting this into context).

So, the key point here is the second part of the justification, to take full advantage of the limited number of positive samples, considering that there are only 58 positive samples (previously identified senolytics).

Hence, there is a trade-off here. If the feature selection process and the 5-fold cross-validation (for selecting the best algorithms and hyperparameter settings) were applied to the 70% training set, rather than the full data, the feature selection and the algorithm/hyperparameter selection would be accessing only about 41 (70% of 58) positive samples. This would make the algorithm/hyperparameter selection somewhat less reliable, by comparison with the current approach of accessing all 58 positive samples. However, the current approach does make the final predictive accuracy results over-optimistic, as discussed above.

To put this trade-off into context, I understand that the main objective of the paper is to use machine learning to identify novel senolytics compounds, using well-established machine learning algorithms, rather than proposing and evaluating new machine learning algorithms or approaches. And it is worth noting that the current experimental methodology led to the discovery of three novel senolytics, including one with improved senolytic performance over a benchmark compound, an interesting result from the perspective of the pharmacology of ageing.

In this context, I accept that the current experimental methodology's goal of making the algorithm/hyperparameter selection more reliable (applying feature selection and cross-validation to the full data) is particularly important, arguably even more important than providing a very precise/'fair' (not over-optimistic) measure of predictive accuracy, even though the latter is obviously also important.

Therefore, although the current experimental methodology is not the ideal from a strict machine learning perspective, in the context of the paper's objective, I accept its use, subject to the condition that the authors explicitly recognise, in the paper, that this methodology is producing over-optimistic results for the final model evaluation on the 30% testing set.

I suggest to add a piece of text like this to the paper:

“It should be noted that the feature selection process accessed the full dataset and the best algorithm and its best hyperparameter settings were selected by applying a 5-fold cross-validation on the full dataset too; and later the same dataset was split into 70% training and 30% testing samples for the final model evaluation. Therefore, the final predictive accuracy results on that set of 30% testing samples are over-optimistic accuracy results, since the selection of the best model benefitted from accessing those 30% testing samples when performing the feature selection and the cross-validation on the full data. In principle these over-optimistic results could be avoided by applying the feature selection process and the 5-fold cross-validation on the 70% training samples, but this would have made the selection of the best model less reliable, since that selection would be made by accessing only about 41 (70% of 58) positive samples. We gave priority to maximise the reliability of the process of best model selection, since the main objective of this paper is to use the best model to identify novel senolytics compounds. However, this methodology has a trade-off: when comparing the final predictive accuracies reported for the best model in this paper against the accuracy of other algorithms applied to our dataset, it should be taken into account that the final predictive accuracies of the best model reported here are over-optimistic, as mentioned above.”

A good place to add this piece of text is at the end of the section titled 'Predicting senolytic compounds by computational screen with machine learning', i.e., right after line 253, so that the above new piece of text will act as a kind of conclusion reflecting on the pros and cons of the experimental methodology from a machine learning perspective.

In addition, when the feature selection process is mentioned in lines 184-185, it is important to make it clear that this used the full dataset.

Hence, in those lines, replace "we first performed a feature importance analysis to decrease the number of features for training"

By "we first performed a feature selection process on the full dataset (before any cross-validation or train-test split), in order to decrease the number of features for training"

Minor points:

A typo in lines 4-5: replace "in both multiple ways, both" by "in multiple ways, both"

Line 213: 'Bayesian classifiers', this is vague, there are several types of Bayesian classifiers, which one are you using? Naïve Bayes? Mention the precise type of Bayesian classifier used here.

Lines 315-317: "More than half of the training compounds were found to be maximally distant from our newly discovered senolytics, which provides strong validation that our machine learning approach can effectively identify diverse compounds..."

In the above sentence, the phrase "strong validation" is an exaggeration. The fact that "more than half training compounds are maximally distant ..." does not tell us anything about how distant the other (less than half) compounds are. Suppose, for example, that a few percent (say 10%) of the training compounds are quite close to the newly discovered senolytics. In this case we could not say that the result would be a strong validation of the diversity of the compounds. I suggest to replace 'strong validation' by a less strong claim, perhaps 'some validation'. More importantly, what mainly matters is not how distant more than half of the training compounds are (to the new discovered senolytics), what mainly matters is the (average) distance between the newly discovered senolytics and its nearest training compounds.

Lines 551-552, 'to optimise the hyperparameter 'tree depth' using a grid search'. It should be mentioned precisely which values of 'tree depth' were considered by the grid search. Lines 554-555 mention that the hyperparameter settings are given in Supplementary Tables 3-5, but these tables mention only the selected settings, not the candidate settings considered by grid search, and it is worth reporting the latter too (only for the best model, XGBoost).

Caption of Supplementary Table 1: replace 'a set of reduced features' by 'a reduced set of features'.

Reviewer #2 (Remarks to the Author):

Having read the revised manuscript and the responses to my comments, I found that the authors have conducted additional experiments and adequately addressed most of my concerns. I, therefore, believe that the manuscript has improved considerably and is now closer to being worthy of publication in Nature Communications. The only disappointment is that these chemicals have not been evaluated in vivo. This point is so important that I wrote about it in my comments at Nature Aging, but it has not been addressed at all, perhaps because the manuscript was transferred to Nature Communications. But even so, the authors should at least mention something about this in the discussion section.

Reviewer #3 (Remarks to the Author):

Major Comments:

This research article focuses on machine learning and the identification of senolytic agents. Here are the major comments and suggestions:

1. This manuscript presents a methodology for identifying senolytic drugs, but it fails to demonstrate its effectiveness.

Among the top 21 compounds (out of 957) predicted to have senolytic action with $P > 44\%$, only three displayed senolytic activity: ginkgetin, oleandrin, and periplocin (as shown in Fig. 3c-d). However, the cell survival assay revealed that only oleandrin could selectively and significantly reduce the survival of senescent cells (as shown in Fig. 4). As a result, the evidence suggests that 1 out of the 21 compounds displays promising senolytic effects. Can the authors please comment on this low true positive hit rate and suggest ways to improve it? Moreover, could this model find those well-known senolytics (such as dasatinib and quercetin)? It is important to know the performance of current senolytics under this model.

2. Is this the first application of machine learning in discovering senolytics? If not, please compare their performance in prediction accuracy and emphasize what the comparative advantages of the current approach are.

It is important to have a section to address the novelty of this approach for identifying senolytic drugs and its significance to the industry. For example, the authors have shown that their ensemble XGBoost model outperformed Chemprop, but they also explained that the difference in the size of training data probably caused it. Also, after retraining the models, the confusion matrices (Fig. 2c) showed that both random forest and XGBoost displayed low false positive rates. However, despite the lower true positive rates by random forest, it yielded 0 false positives compared to 4 by the XGBoost model. As the authors also mentioned in the text (line 217-219), false positives are more deleterious than false negatives, did the authors employ random forest to screen the library of chemical structures to see whether it could also prioritize Oleandrin or even better senolytics? How would the authors convince the audience that their model and approach are superior and could deliver a significant impact on compound screening with machine learning?

3. Senescent cells exhibit a phenotype known as senescence-associated secretory phenotype (SASP), wherein they secrete high levels of inflammatory cytokines, immune modulators, growth factors, and proteases. It is worthwhile to investigate the effects of senolytic drugs on SASP, as well as the effects of senescent cell removal by senolytic agents. Several in vitro assays can be used to investigate the effects of senolytic agents on senescent cell removal. For example, senescence-associated beta-galactosidase (SA-beta-gal) assay, detection of senescent cells using antibodies against senescence markers such as p16 and p21.

Response to reviewers

“Discovery of senolytics using machine learning”.

Reviewer #1

Q: *This revised version of the paper has a clearer and more detailed description of the machine learning methodology, by comparison with the original submission; and the authors have provided detailed responses to my previous comments.*

A: Thank you for the detailed and careful assessment of our work; your comments have been incredibly helpful.

The reviewer provided a detailed and well-argued analysis about our use of a 70:30 split for evaluating the performance of the final models. Their main point is that the reported metrics, trained on 70% of the data and evaluated on the held-out 30%, are over-optimistic due to leakage in feature selection and k-fold cross validation.

We agree with this point and appreciate that the Reviewer accepts our choice on the basis that it allowed us to make maximal use of the few positives we had for training. The reviewer is right in pointing out that this decision is non-standard from a machine learning perspective, and we have included a paragraph to clarify the above (L254-265). We have slightly edited the reviewers' suggested paragraph for conciseness:

“We note that our feature selection process accessed the full dataset, and the best algorithm and hyperparameter settings were selected by applying a 5-fold cross-validation on the full dataset, too. Therefore, the performance metrics computed on a set of 30% held-out compounds are over-optimistic, since model selection benefitted from accessing those testing samples during feature selection and cross-validation on the full data. In principle, these over-optimistic results could be avoided by applying the feature selection and the 5-fold cross-validation on the 70% training samples, but this would have reduced the training set to only 41 positive compounds, and thus make the selection of the best model less reliable. Since our aim was to use the best model to screen for new senolytic compounds, we prioritized a reliable selection of the best model for screening. The caveat of this approach is that it leads to over-optimistic performance metrics for the best models when compared to other algorithms applied to our dataset.”

Q: *In addition, when the feature selection process is mentioned in lines 184-185, it is important to make it clear that this used the full dataset. Hence, in those lines, replace “we first performed a feature importance analysis to decrease the number of features for training” By “we first performed a feature selection process on the full dataset (before any cross-validation or train-test split), in order to decrease the number of features for training”.*

A: We agree and have included the suggested edit in L183-185.

Q: *A typo in lines 4-5: replace “in both multiple ways, both” by “in multiple ways, both”.*

A: This has been corrected.

Q: *Line 213: ‘Bayesian classifiers’, this is vague, there are several types of Bayesian classifiers, which one are you using? Naïve Bayes? Mention the precise type of Bayesian classifier used here.*

A: We apologise for the omission; we indeed employed Naïve Bayes and have clarified this in the text.

Q: Lines 315-317: “More than half of the training compounds were found to be maximally distant from our newly discovered senolytics, which provides strong validation that our machine learning approach can effectively identify diverse compounds...”

In the above sentence, the phrase “strong validation” is an exaggeration. The fact that “more than half training compounds are maximally distant ...” does not tell us anything about how distant the other (less than half) compounds are. Suppose, for example, that a few percent (say 10%) of the training compounds are quite close to the newly discovered senolytics. In this case we could not say that the result would be a strong validation of the diversity of the compounds. I suggest to replace ‘strong validation’ by a less strong claim, perhaps ‘some validation’. More importantly, what mainly matters is not how distant more than half of the training compounds are (to the new discovered senolytics), what mainly matters is the (average) distance between the newly discovered senolytics and its nearest training compounds.

A: Thank you for this insightful comment; we agree and have edited the claim in L328.

Q: Lines 551-552, ‘to optimise the hyperparameter ‘tree depth’ using a grid search’. It should be mentioned precisely which values of ‘tree depth’ were considered by the grid search. Lines 554-555 mention that the hyperparameter settings are given in Supplementary Tables 3-5, but these tables mention only the selected settings, not the candidate settings considered by grid search, and it is worth reporting the latter too (only for the best model, XGBoost).

A: This has been corrected.

Q: Caption of Supplementary Table 1: replace ‘a set of reduced features’ by ‘a reduced set of features’.

A: This has been corrected.

Reviewer #2

Q: *Having read the revised manuscript and the responses to my comments, I found that the authors have conducted additional experiments and adequately addressed most of my concerns. I, therefore, believe that the manuscript has improved considerably and is now closer to being worthy of publication in Nature Communications. The only disappointment is that these chemicals have not been evaluated in vivo. This point is so important that I wrote about it in my comments at Nature Aging, but it has not been addressed at all, perhaps because the manuscript was transferred to Nature Communications. But even so, the authors should at least mention something about this in the discussion section.*

A: Thank you for the encouraging assessment of our work. We have added a comment on *in vivo* tests in the Discussion section (L464-465), as suggested.

Reviewer #3

Q: *This manuscript presents a methodology for identifying senolytic drugs, but it fails to demonstrate its effectiveness. Among the top 21 compounds (out of 957) predicted to have senolytic action with $P > 44\%$, only three displayed senolytic activity: ginkgetin, oleandrin, and periplocin (as shown in Fig. 3c-d). However, the cell survival assay revealed that only oleandrin could selectively and significantly reduce the survival of senescent cells (as shown in Fig. 4). As a result, the evidence suggests that 1 out of the 21 compounds displays promising senolytic effects. Can the authors please comment on this low true positive hit rate and suggest ways to improve it?*

A: Thank you for the comments. There are several precisions to be made:

- The dose-response curves shown in Fig. 3c,d already show that three compounds (oleandrin, periplocin, ginkgetin) selectively target oncogene-induced senescent cells over normal cells. Supplementary Figure 6d shows similar senolytic action of the three compounds on therapy-induced senescent cells.

The survival assay mentioned by the Reviewer (Fig. 4a,b) was done solely with the purpose of demonstrating the higher potency of oleandrin against a well-known senolytic (ouabain) and one of the other two hits (periplocin). Therefore, the assay was deliberately done at a low concentration for the three compounds (10nM). The assay in Fig. 4a,b therefore does not imply that 'only oleandrin could selectively and significantly reduce the survival of senescent cells' as suggested.

- The model could indeed successfully predict 3 out of 21 experimentally screened compounds, as shown in Fig. 3c-d. This is not a low true positive rate. It is, in fact, a strong result because it amounts to a discovery rate of DR=14.3%, which is much higher than previous senolytic screens such as Guerrero et al, Nat Metab, 2019 (DR=2.5%), Triana-Martinez et al, Nat Commun, 2019 (DR=0.4%), and Wakita et al, Nat Commun, 2020 (DR<0.01%).
- We computationally screened 4,340 compounds, not 957. This is mentioned in the main text and methods. We have provided the list of compounds screened in Zenodo at <https://doi.org/10.5281/zenodo.7870357>.

Q: *Moreover, could this model find those well-known senolytics (such as dasatinib and quercetin)? It is important to know the performance of current senolytics under this model.*

A: We believe there is a confusion here. The models were trained on 58 senolytics, including dasatinib and quercetin (see Tanimoto graph in Figure 1d). Therefore, if we query the models for these two senolytics, it will return high prediction scores solely because both compounds were in the training set. Such high prediction score would not reflect the quality of the model; assessing model quality on the training data is considered against best practices in machine learning. For realistic performance assessment, evaluation of machine learning models must be done on test data that was not employed for training, for example through k-fold cross-validation as we did in Fig. 2B and Supplementary Table 2.

Q: *Is this the first application of machine learning in discovering senolytics? If not, please compare their performance in prediction accuracy and emphasize what the comparative advantages of the current approach are. It is important to have a section to address the novelty of this approach for identifying senolytic drugs and its significance to the industry.*

A: To the best of our knowledge, our work is the first to report the discovery of senolytics with machine learning algorithms trained on published data, and thus large reductions in experimental screening costs. We have extensively discussed the innovations of our work in L399-413 of the Discussion, and the other referees have explicitly acknowledged the novelty of the approach.

Quantitative comparisons between our XGBoost model and other models are difficult to perform because existing algorithms differ vastly on the type of data employed for training. In problems with small and heterogeneous data, quantitative comparisons across models must be done with caution, because models have been heavily optimised to the question and data at hand. There are important design decisions that are bespoke to the task, such as the input data, its featurization strategy, and the choice of machine learning models. For example, Kusumoto et al, Nature Communications, 2021 (ref [48] in the manuscript), found compounds with senolytic action using microscopy imaging data and deep learning algorithm specialized for image recognition. Our XGBoost model, in contrast, was designed to process data in the form of physicochemical descriptors, and cannot process imaging data without major modifications, which would make such comparisons of little value.

Q: *For example, the authors have shown that their ensemble XGBoost model outperformed Chemprop, but they also explained that the difference in the size of training data probably caused it.*

A: The comparison with chemprop was done to justify our model design decisions and we do not mean to imply that our approach is generally better. It is likely that in drug discovery tasks with a larger number of positives for training, state-of-the-art deep learning such as chemprop, graph transformers (Rong et al, arXiv, 2020) or the newly developed large language models (Ahmad et al, arXiv, 2022) could produce strong results.

But in the case of senolytics, there is simply not enough data for training accurate deep learning models. We thus had to work with this limited dataset and devise a way of extracting useful information from it. We found that XGBoost was the best model for this task, and our experimental hit validation in two cell lines and three modes of senescence induction confirm the success of our strategy.

Q: *Also, after retraining the models, the confusion matrices (Fig. 2c) showed that both random forest and XGBoost displayed low false positive rates. However, despite the lower true positive rates by random forest, it yielded 0 false positives compared to 4 by the XGBoost model. As the authors also mentioned in the text (line 217-219), false positives are more deleterious than false negatives, did the authors employ random forest to screen the library of chemical structures to see whether it could also prioritize Oleandrin or even better senolytics?*

A: We agree that the confusion matrices in Fig. 2C show that the random forest has a lower false positive rate than XGBoost, but these matrices correspond to just one evaluation run and thus it is not statistically representative. Assessing model performance on a single run often leads to overoptimistic results and this is why the best practice is to evaluate models across several randomized validation sets. As we show in Fig. 2B and Supplementary Table 2, the 5-fold cross-validation results clearly show that Random Forest model is more prone to overfitting than the XGBoost; this is reflected by the large standard deviation in the performance metrics. As a result, we discarded the Random Forest model and did not employ it for screening.

Q: *How would the authors convince the audience that their model and approach are superior and could deliver a significant impact on compound screening with machine learning?*

A: Our contribution is not 'a superior model and approach' as suggested, but rather the discovery and validation of three senolytics, a small but important class of compounds for which there are few molecular targets known. To achieve this, we devised a bespoke machine learning pipeline trained on the few screening data available for senolytics. The success of our approach demonstrates that careful data curation/analysis coupled with tailored machine learning algorithms can effectively aid early-stage drug discovery, even in cases where existing data is highly limited and heterogeneous.

Q: *Senescent cells exhibit a phenotype known as senescence-associated secretory phenotype (SASP), wherein they secrete high levels of inflammatory cytokines, immune modulators, growth factors, and proteases. It is worthwhile to investigate the effects of senolytic drugs on SASP, as well as the effects of senescent cell removal by senolytic agents. Several in vitro assays can be used to investigate the effects of senolytic agents on senescent cell removal. For example, senescence-associated beta-galactosidase (SA-beta-gal) assay, detection of senescent cells using antibodies against senescence markers such as p16 and p21.*

A: Thank you for this important point. We fully agree on the importance of the effect of senolytics on the SASP and the senescence phenotype itself. In the revision we conducted further experiments to investigate how one of our newly discovered senolytics, oleandrin, affected two senescence markers (p21, p16) and three SASP components (IL1 α , IL1 β , and IL8) in surviving cells following treatment.

Our findings show that:

- After oleandrin treatment, surviving cells displayed low expression of p21 and p16, confirming the elimination of most senescent cells in the culture; this is shown in the new Supplementary Figure 9j-k for oncogene-induced and replicative senescence, respectively.
- Oleandrin at low concentrations reduced the expression of IL1 α , IL1 β and IL8, suggesting that oleandrin eliminated most SASP-expressing cells; this is shown in the new Supplementary Figure 9l-m for oncogene-induced and replicative senescence, respectively.

These results emphasize that oleandrin is effective in eliminating senescent cells, and that surviving cells after oleandrin treatment do not retain the proinflammatory program. This is significant because the SASP is believed to be responsible for the deleterious effects of senescent cells in pathology, and the ultimate goal of using senolytics is to remove SASP-expressing cells from tissues. We have discussed these new results in L382-390 of the main text.

REVIEWERS' COMMENTS

Reviewer #3 (Remarks to the Author):

The authors revised the manuscript with additional experiments to address my main concern. I'm also fine with their responses to my comments.

Response to reviewers

“Discovery of senolytics using machine learning”.

Reviewer #3

Q: *The authors revised the manuscript with additional experiments to address my main concern. I'm also fine with their responses to my comments.*

A: Thank you for your comments and helping us improve our manuscript.